# Phenolic Constituents, Antioxidant and Cytoprotective Activities, Enzyme Inhibition Abilities of Five Fractions from *Vaccinium dunalianum* Wight

**DOI:** 10.3390/molecules27113432

**Published:** 2022-05-26

**Authors:** Chang-Shu Cheng, Qing-Hui Gu, Jin-Ke Zhang, Jun-Hong Tao, Tian-Rui Zhao, Jian-Xin Cao, Gui-Guang Cheng, Guo-Fang Lai, Ya-Ping Liu

**Affiliations:** 1Faculty of Food Science and Engineering, Kunming University of Science and Technology, Kunming 650500, China; chengchangshu0409@163.com (C.-S.C.); gu18082917744@163.com (Q.-H.G.); z17853480162@163.com (J.-K.Z.); tjh15198526834@163.com (J.-H.T.); food363@163.com (T.-R.Z.); jxcao321@hotmail.com (J.-X.C.); ggcheng@kust.edu.cn (G.-G.C.); 2Yunnan Institute for Food and Drug Control, Kunming 650106, China

**Keywords:** *Vaccinium dunalianum* Wight, chemical constituents, enzyme inhibition, antioxidant capacity, oxidative damage protection

## Abstract

The bud of *Vaccinium dunalianum* Wight has been traditionally consumed as health herbal tea by “Yi” people in Yunnan Province, China, which was locally named “Que Zui tea”. This paper studied the chemical constituents of five fractions from *Vaccinium dunalianum*, and their enzyme inhibitory effects of α-glucosidase and pancreatic lipase, antioxidant activity, and cytoprotective effects on H_2_O_2_-induced oxidative damage in HepG2 cells. The methanol extract of *V. dunalianum* was successively partitioned with petroleum ether (PF), chloroform (CF), ethyl acetate (EF), *n*-butanol (BF), and aqueous (WF) to obtain five fractions. The chemical profiling of the five fractions was analyzed by ultra-high-performance liquid chromatography coupled with a tandem mass spectrometry (UHPLC-MS/MS), and 18 compounds were tentatively identified. Compared to PF, CF, BF and WF, the EF revealed the highest total phenols (TPC) and total flavonoids (TFC), and displayed the strongest enzyme inhibition ability (α-glucosidase and pancreatic lipase) and antioxidant capacity (DPPH, ABTS and FRAP). Furthermore, these five fractions, especially EF, could effectively inhibit reactive oxygen species (ROS) production and cell apoptosis on H_2_O_2_-induced oxidative damage protection in HepG2 cells. This inhibitory effect might be caused by the up-regulation of intracellular antioxidant enzyme activity (CAT, SOD, and GSH). The flavonoids and phenolic acids of *V. dunalianum* might be the bioactive substances responsible for enzyme inhibitory, antioxidant, and cytoprotective activities.

## 1. Introduction

Reactive oxygen species (ROS) are considered to be one of the most important endogenous free radicals, created during normal cellular metabolism through peroxidation respiration. ROS generated by cells as products or byproducts, can function either as signaling mols [1]. However, when the body is subjected to adverse external stimuli, excess ROS will be produced [2]. In the event that the ROS cannot be really eliminated, excessive ROS would generate oxidative stress, which could affect cell multiplication and apoptosis [3]. In addition, the main source of free radicals, oxidative damage, is not only the root of chronic inflammation, type 2 diabetes, cardiovascular disease, and many other chronic diseases, but also a risk factor closely related to metabolic dysfunction [4]. Therefore, many strategies are needed to deal with the excess ROS. Furthermore, finding effective and safe bioactive molecules from nature to scavenge excessive ROS is critical, with phenolic compounds taking center stage [5]. 

Oxidative stress refers to a state of imbalance between oxidation and anti-oxidation in the body; a vast number of oxidation intermediates produce free radicals in the body, creating oxidative stress, which is a detrimental consequence that is thought to play a role in diseases, such as type 2 diabetes, neurodegenerative diseases, and obesity [6]. Oxidative stress has been reported to decrease insulin gene expression and it can even harm the islet [7], α-glucosidase is a critical enzyme in the digestion and absorption of carbs [8], which is one of the main members of the body responsible for glucose metabolism enroute [9]. As it is well-known that pancreatic lipase is a critical enzyme in the breakdown of triglycerides in the gastrointestinal tract [10]. Thus, improving oxidative stress can reduce the incidence of diseases like diabetes and obesity. Phenolic compounds are important chemical constituents from natural resources, which are the products of the secondary metabolism of natural plants. Many studies have reported that phenolic compounds have a good effect on improving oxidative stress, and have anti-inflammatory and anti-aging properties [11,12]. A great number of studies found that polyphenols had good antioxidant activity and could be used as natural antioxidants, which can help the body scavenge free radicals and prevent the physical damage caused by oxidative stress [13].

The bud of *Vaccinium dunalianum* Wight (*V. dunalianum*) has traditionally been used as health herbal tea by “Yi” people in Yunnan Province, China. Because its shape resembles a sparrow’s mouth, it was locally named “Que Zui tea” (Figure 1) [4]. Hyperlipidemia, diabetes, rheumatoid arthritis, and depression-like symptoms can all be prevented by long-term ingestion of *V. dunalianum* [14]. Studies on *V. dunalianum* revealed caffeic acid derivatives, flavonoids, and arbutin derivatives were the main components [15], and these compounds displayed many activities, such as anti-inflammatory [16], antibacterial [17], anticancer [18], and neuroprotective activities [19]. However, the enzyme inhibition activities and antioxidants caused by oxidative damage of *V. dunalianum* were seldom revealed. The current research evaluated the enzyme inhibitory activity, antioxidant activity, and cytoprotective activity against H_2_O_2_-induced oxidative damage in HepG2 cells of five fractions from *V. dunalianum*. These active substances were also analyzed by ultra-performance liquid chromatography-mass spectrometry (UHPLC-MS/MS), and 18 compounds were tentatively identified. The EF revealed the highest TPC and TFC, and displayed the strongest enzyme inhibition ability (α-glucosidase and pancreatic lipase) and antioxidant capacity (DPPH, ABTS and FRAP). Furthermore, EF could effectively inhibit ROS production and cell apoptosis on H_2_O_2_-induced oxidative damage protection in HepG2 cells, which might be caused by up-regulation of intracellular antioxidant enzyme activity (CAT, SOD and GSH). This study provides scientific data for the future application of *V. dunalianum* as dietary supplements with human benefits in functional foods.

## 2. Results and Discussion

### 2.1. Total Phenolic Content and Total Flavonoid Content

Phenolic compounds are important secondary metabolites in natural plants. As well, this class of compounds can protect the organism from infection [20]. Numerous studies have shown that phenolics have anti-tumor [21], hypoglycemic [22], and hypolipidemic [23] functions. They also have strong antioxidant properties, which might be the pharmacological basis for other biological effects [6]. In this study, the extract of *V. dunalianum* was divided into five fractions by organic reagents to enrich the different types of compounds. The TPC and TFC of five different fractions from *V. dunalianum* were detected by spectrophotometric assays, and are shown in Table 1. The TPC was measured in gallic acid equivalents, while TFC was measured in rutin equivalents. Furthermore, the TPC and TFC of the five fractions were found to be considerably different (*p* < 0.05) (Table 1). As a result, the EF displayed the highest TPC values of 59.65 ± 0.93 mg GAE equivalent/g extract, while the PF showed the lowest TPC values of 12.68 ± 0.37 mg GAE equivalent/g extract (*p* < 0.05), these data suggested the TPC of EF was about five times higher than that of PF. Like the overall trend of TFC of five fractions, EF had the highest content of TFC, followed by the BF, CF, WF, and PF. In addition, the TFC of EF was approximately four times higher than that of PF, and the difference in the degree of the TFC of five fractions was less than that of TPC.

### 2.2. Chemical Composition Analysis of Five Fractions

The chemical compositions of the five fractions from *V. dunalianum* were performed by UHPLC-MS/MS analysis. The compounds were identified and quantified based on standards, references, and the Massbank database. The details of identified compounds in PF, CF, BF, EF and WF were generalized and described in Table 2, including M/Z values, retention times, molecular formulae, and MS/MS fragments. The total ion chromatograms of five fractions were displayed in Figure 2. A total of 18 compounds, including 6 flavonoids and 12 phenolic acids, were detected with standards and reported data in the five fractions (Table 2). As displayed in Figure 2 and Table 2, a total of 18 compounds were detected in CF, EF, and BF, while 14 compounds were detected in WF with the absence of compounds **7**, **8**, **9,** and **15**. Moreover, four compounds were detected in substance PF (compounds **1**, **2**, **5**, and **14**), while compounds **3**, **4**, **6**, **7**, **8**, **9**, **10**, **11**, **12**, **13**, **15**, **16**, **17**, **18** were not detected. Compounds **1**, **4**, **6**, and **16** were queried by using the Massbank database and chemicals **2** and **17** were semi-quantified by protocatechuic acid. While compounds **4**, **7**, and **14** were semi-quantified by chlorogenic acid, compound **18** was semi-quantified by kaempferol. The remaining chemicals were determined based on literature reports. 

Under the same conditions, the corresponding standards and standard curves were established based on the different concentrations of the standards and the corresponding peak areas using UHPLC-MS/MS analysis. According to Figure 2, it is obvious that the peak areas and heights of most of the compounds were greatly different, which was significantly influenced by the organic solvent distribution. After further successively partitioning with petroleum ether, chloroform, ethyl acetate, and *n*-butanol, the compounds were partitioned based on the principle of similarity-intermiscibility. The content of most of the compounds was more evidently increased in the EF than that of CF and BF, especially that of PF and WF. The EF showed the highest number and content of compounds, while the PF displayed the lowest. The concentration of each compound was calculated by combining the peak areas and presented in Table 3, the quinic acid (**1**), chlorogenic acid, (**4**) and 6’-O-caffeoylarbutin (**14**) were the major phenolic compounds. The content of quinic acid (**1**) of WF, CF and BF was 239,190.25 ± 39.42, 22,049.23 ± 49.02 and 19,221.03 ± 21.05 μg/g dry extract, individually; about 190, 17 and 15 times to that of EF (1253.7 ± 3.05 μg/g dry extract). The content of chlorogenic acid (**4**) in BF was 129,305.52 ± 102.23 μg/g dry extract, while in WF and EF was 47,967.83 ± 67.51 and 42,713.84 ± 12.23 μg/g dry extract respectively, and CF was 25,730.28 ± 32.05 μg/g dry extract. The content of BF was about three, three, and five times those of WF, EF and CF, respectively. The contents of 6’-O-caffeoylarbutin (**14**) in EF and BF were 679,611.55 ± 17.56 and 387,775.65 ± 61.99 μg/g dry extract, individually, about 24,271 and 13,849 times that of PF (28.27 ± 2.53 μg/g dry) (Table 3). These phenomena clearly indicated that the use of solvents to purify and enhance phenolic compounds is a successful strategy. Because of their solubility and the polarity of the extraction solvent, the compounds were enriched.

### 2.3. Enzyme Inhibition Activity of Five Fractions

#### 2.3.1. α-Glucosidase Inhibitory Activity

The activity of α-glucosidase has been linked to a number of illnesses, including diabetes [24]. The results of α-glucosidase inhibitory activities of five fractions were shown in Table 1, the EF showed better inhibitory activity against α-glucosidase with an IC_50_ value of 84.21 ± 2.21 μg/mL than those of the other four fractions, which was significantly different from the other four fractions (*p* < 0.05). The PF showed the worst inhibitory effect (IC_50_ = 498.56 μg/mL), which IC_50_ value was about six times that of EF. The orders of inhibitory activities against α-glucosidase were as follows EF > BF > CF > WF > PF. Based on the determination of TPC and TFC contents, EF contained the most TPC and TFC, while PF contained the least TPC and TFC, these results suggested that α-glucosidase inhibitors were consistent with the content of phenolic components of the five fractions from *V. dunalianum*. As well, according to many studies [4,18], the higher the content of TPC and TFC, the higher the α-glucosidase inhibiting activities.

#### 2.3.2. Inhibitory Activity of Pancreatic Lipase

For pancreatic lipase, its main matter is the enzyme of hydrolyzes fats [25]. Table 1 shows the pancreatic lipase inhibitory activity of five fractions, the EF had the best pancreatic lipase inhibitory activity (IC_50_ = 122.53 ± 5.45 μg/mL), which was substantially different from the other groups (*p* < 0.05). The PF had the worst inhibitory effect (IC_50_ = 814.32 ± 6.53 μg/mL), with IC_50_ values of about two, three, four, and seven times those of WF, CF, BF, and EF, respectively. The order of inhibitory activity of different fractions against pancreatic lipase was the same as the order of α-glucosidase inhibiting activities. Duo to the determination of TPC and TFC contents, the EF contained the most TPC and TFC, which indicated that the phenolic components of the five fractions from *V. dunalianum* were the main pancreatic lipase inhibitors. The higher TPC and TFC contents, the stronger the inhibition activity of pancreatic lipase.

#### 2.3.3. Antioxidant Effects of Five Fractions

Many studies reported that phenolic compounds have good antioxidant activity, which playedan important role in the health of the organism [4,26,27]. A number of flavonoids and phenolic acids have been developed as popular antioxidant foods (nutritional products), which were beneficial to human health [28]. The antioxidant activities of the five fractions were evaluated by three general methods (FRAP, ABTS, and DPPH). The ABTS, DPPH, and FRAP values were expressed as nmol trolox equivalent (TE)/g extract, and is displayed in Table 1. Among these five fractions, EF had the best radical scavenging activities for FRAP (0.91 ± 0.04 nmol TE/g extract), ABTS (12.85 ± 0.23 nmol TE/g extract), and DPPH (0.44 ± 0.03 nmol TE/g extract), respectively, followed by BF. PF showed no antioxidant activity in DPPH, ABTS radical scavenging as well as the lowest FRAP antioxidant capacity with 1.35 ± 0.14 nmol TE/g extract (Table 1). Depending on the content of TPC and TFC of each fraction, the antioxidant capacities were changed with the contents. Furthermore, based on their variations, it can be inferred that the antioxidant capacities of five fractions were correlated with the content of TPC and TFC (DPPH values r = 0.91 and 0.93; FRAP values r = 0.89 and 0.86); but ABTS had poor correlation with TPC and TFC, with correlation r = 0.55 and 0.63, respectively. These suggested that there was a strong association with DPPH and FRAP of TPC and TFC, and a moderate correlation between ABTS and TPC and TFC. In general, ABTS, DPPH, and FRAP were correlated with TPC and TFC. These data indicated that there was a certain relationship between free radical scavenging ability and TPC and TFC.

### 2.4. Cytoprotective Effect on Oxidative Damage in H_2_O_2_-Induced HepG2 Cells

#### 2.4.1. Inhibitory Effect on Intracellular Reactive Oxygen (ROS) Generation

HepG2 cells have been used regularly to build model organisms for biomedical exploration, like cytoprotective effects and ROS [29]. HepG2 cells have many advantages, including easy culture, rapid propagation, and cheapness compared to normal cells. ROS are involved in cell signal transduction, oxidative stress induction, nucleic acid and protein damage, etc. Abnormal ROS can cause abnormal physiological functions and diseases [6]. More ROS produced can lead to lipid and protein oxidation stress, cell damage and aging, and even apoptosis [24]. Generally, H_2_O_2_ can induce the abnormal accumulation of ROS in cells and break the antioxidant defense ability of cells, which was used to construct the cell oxidative damage model in vitro [25,26]. The MTT test uncovered that the five fractions indicated non-toxic to HepG2 cells at 150 μg/mL concentration. The concentrations of 50, 100, and 150 μg/mL of each fraction were chosen to assess ROS scavenging ability. The inhibitory effect of five fractions from *V. dunalianum* was evaluated on intracellular ROS generation in H_2_O_2_-induced HepG2 cells, and the levels of intracellular ROS were tested by flow cytometry. H_2_O_2_ significantly caused the accumulation of ROS in HepG2 cells compared with the control group (*p* < 0.05), indicating that the H_2_O_2_-induced oxidative damage model in HepG2 cells was successfully established (Figure 3). The Vc, used as a positive control, significantly inhibited the accumulation of ROS in HepG2 cells compared to the H_2_O_2_ group (*p* < 0.05). All five fractions of *V. dunalianum* significantly inhibited the accumulation of cells’ ROS in a dose-dependent manner (*p* < 0.05) (Figure 3). Among the EF displayed the strongest suppressive effect on intracellular ROS production (*p* < 0.05), which was equal to the Vc group at the concentrations of 150 μg/mL (Figure 3). Overall, the abilities of five fractions to inhibit intracellular ROS production: EF > BF > CF > WF > PF.

Phenols exhibited strong antioxidant capacities by inhibiting the production of intracellular ROS induced by H_2_O_2_ [4]. In this experiment, the cell ROS inhibitory effect of the five fractions was highly correlated with the contents of TPC and TFC. Among them, EF contained the most TPC and TFC, and exhibited the best ROS inhibition effect. Thus, the phenols and flavonoids in *V. dunalianum* played an important role in the inhibition of intracellular ROS production.

#### 2.4.2. Effect on Intracellular Antioxidant Enzyme Activities

Endogenous antioxidant enzymes including superoxide dismutase (SOD), catalase (CAT), and glutathione (GSH) play an important role in the prevention of oxidative stress [30]. Moreover, it is well known that SOD catalyzed the disproportionation of superoxide anion radicals to generate oxygen and hydrogen peroxide, which played a crucial role in the oxidative and antioxidant balance of the body and are inextricably linked to the onset and development of many diseases [31]. GSH is a cellular antioxidant molecule that scavenges H_2_O_2_, and it has been shown to enhance superoxide dismutase, hydrogen peroxidase, and glutathione peroxidase levels, which could improve human health and lead to a longer life span [32].

By measuring SOD, CAT and GSH, determining the enzymatic antioxidant activity of five fractions from *V. dunalianum* against H_2_O_2_-induced HepG2 cells. In the current research, the activities of GSH, CAT and SOD essentially diminished in the H_2_O_2_-induced group than the control group (*p* < 0.05) (Figure 4). Within the sight of 1.0 mM H_2_O_2_, the treatment groups with five fractions remarkably increased the SOD enzymatic activities (Figure 4A), among which EF had the best activity, trailed by BF and CF, while the PF, WF exhibited the lowest activity. The EF displayed the highest SOD levels by increasing to 100% at the concentrations of 150 μg/mL, which was equivalent to the Vc group and control group (*p* < 0.05) (Figure 4A). The EF displayed the highest CAT levels at the concentrations of 150 μg/mL, which was equivalent to the Vc group, while the PF had the lowest enzymatic activity (Figure 4B). In addition, EF had a good effect on GSH activity, while WF had a poor activity (Figure 4C). The information confirmed that the enzymatic activities of SOD, GSH, and CAT were significantly down controlled by H_2_O_2_ treatment, and five fractions from *V. dunalianum* remarkably increased these antioxidant enzymatic activities, and the concentrations dependent manner on GSH, SOD, and CAT. 

#### 2.4.3. Cytoprotective Activity against H_2_O_2_-Induced Cell Apoptosis

Cell apoptosis is a fundamental natural peculiarity of cells, which assumes a significant part in the administrative system of cells’ proliferation, development, and mutation, and the strength of the internal environment [33,34]. However, abnormal apoptosis, caused by oxidative stress, can lead to autoimmune and neurodegenerative diseases [35]. H_2_O_2_ is considered as a significant molecule in signal transduction pathways since it manages different cell capacities, for example, apoptosis, proliferation, and growth arrest [36]. The present study monitored the apoptosis of HepG2 cells induced by H_2_O_2_ and evaluated the cells’ protection effect of five fractions from *V. dunalianum* against H_2_O_2_-induced apoptosis. The non-toxic dose of cells was screened by MTT assay. The apoptosis ratio (48.46 ± 4.38%) significantly increased in HepG2 cells by 1.0 mM H_2_O_2_, in comparison with the control (1.76 ± 0.16%) (*p* < 0.05) (Figure 5). Nonetheless, the level of apoptotic cells treated by five fractions from *V. dunalianum* was diminished drastically when contrasted with that of the H_2_O_2_-treated group (*p* < 0.05). For those, the EF and BF exhibited the strongest cytoprotective effect with apoptosis ratios of 3.31% and 3.58%, respectively, which were equal to the Vc group with an apoptosis ratio of 3.17%, while PF had the lowest activity with an apoptosis ratio of 13.67% (*p* < 0.05). These revealed that *V. dunalianum* significantly inhibited H_2_O_2_ induced cell apoptosis, especially EF and BF, these were possibly the difference substance of flavonoids and phenolics in difference fractions. On the basis of TPC, TFC, and cell apoptosis proportion of five fractions, the cell apoptosis rate was shown to be negatively correlated with the TPC (r = 0.90) and TFC (r = 0.91), implying that the phenolic and flavonoid content of *V. dunalianum* contributed significantly to the cytoprotective effect.

The antioxidant effects of the five fractions from *V. dunalianum* were the same trend as their cell apoptosis effects, the same as intracellular ROS effects on H_2_O_2_-induced HepG2, and the same as on the scavenging ability of DPPH and ABTS free radicals and FRAP results. These data indicated that *V. dunalianum* has a good antioxidant effect and can reduce oxidative damage in cells by reducing the generation of intracellular ROS and free radicals. Our studies showed that *V. dunalianum* has a protective effect against H_2_O_2_-induced oxidative damage in PC12 cells, which can significantly reduce the production of intracellular ROS and improve the survival rate of damaged cells [19]. Compared with the leave, fruit of *V. dunalianum*, the bud of *V. dunalianum* displayed the strongest antioxidant capacity and protective effect [19]. These results displayed that the EF fraction had the best antioxidant effect, with the highest content of TPC and TFC. In conclusion, further research can be conducted to develop the potential of *V. dunalianum* for use as a natural antioxidant.

### 2.5. Multivariate Analysis

As seen in Figure 6, this principal component analysis (PCA) explained 97.8% of the total variance in the stomach, 88.9% of the variance explained by PC1, and 8.9% of the variance explained by PC2. The EF and BF were distributed in the PC1 on the upper right quadrant (positive side), which contained high levels of TPC, TFC, GSH, SOD, and in vitro antioxidant indexes; while CF, WF, and PF were placed in the left side of PC1 (negative test), which have low levels of TPC, TFC, GSH, SOD, and in vitro antioxidant indexes. Therefore, for the five fractions of *V. dunalianum*, the high content of TPC and TFC has a great influence on their antioxidant capacity.

## 3. Materials and Methods

### 3.1. Materials and Reagents

Sodium hydroxide, methanol, ethanol, petroleum ether, chloroform, acetone, and ethyl acetate are from Chongqing Chuandong Chemical (Group) Co. LTD (Chongqing, China), mass spectrometry grade acetonitrile, formic acid (Merck, Germany); standards (Manster, Chengdu, China); α-glucosidase, porcine pancreatic lipase, acarbose, orlistat, DMEM (Dulbecco’s Modified Eagle Medium) medium, Phosphate buffered saline (PBS), fetal bovine serum (Gibco, New York, NY, USA); 3-(4,5-dimethylthiazol-2)-2,5-diphenyltetrazolium bromide salt (MTT), DCFH-DA Dimethyl sulfoxide (DMSO) and trypsin (Sigma Chemical Company, Burlington, MA, USA); superoxide dismutase (SOD) kit, glutathione peroxidase (GSH) kit and catalase (CAT) kit (Nanjing Jiancheng Institute of Biological Engineering; Nanjing, China); apoptosis kit (Beijing Si Zhengbai Biotechnology Co.; Beijing, China)

### 3.2. Plant Material

The buds of *Vaccinium dunalianum* Wight were collected from Chuxiong Yi Autonomous Prefecture, Yunnan Province in mid-June, 2016, and were identified by Prof. Jian-Xin Cao, Faculty of Food Science and Engineering, Kunming University of Science and Technology (KUST). The sample (No. Cao20160609) was stored in the laboratory of the Faculty of Food Science and Engineering, KUST.

### 3.3. Extraction and Preparation of Five Fractions of Vaccinium Dunalianum

The collected buds of *V. dunalianum* were dried in a shade room with great ventilation at 20 ± 5 °C until they reached a constant weight. The dried material (2 kg) was ground into powder and sieved through a 60-mesh sieve before being extracted three times with 80% methanol-water (5 L) using ultrasonic aided extraction (30 min each time). The suspension was collected and centrifuged at 4000× *g* for 10 min to obtain the supernatant. Then, the supernatant was concentrated by rotary evaporation device (Heidolph, Schwabach, Germany). Then, the extract was successively partitioned with petroleum ether, chloroform, ethyl acetate and *n*-butanol. Firstly, the extract was partitioned with petroleum ether to afford petroleum ether fraction and water fraction. Secondly, the water layer was partitioned with chloroform to give chloroform fraction and water fraction. Thirdly, the water fraction was extracted with ethyl acetate and then extracted with *n*-butanol to give ethyl acetate fraction, *n*-butanol fraction and last water fraction. Each solvent was extracted 3 times. Last, petroleum ether fraction (PF), chloroform fraction (CF), ethyl acetate fraction (EF), *n*-butanol fraction (BF) and water fraction (WF) were obtained by concentration. The five fractions were obtained by partitioning the extract with five different solvents successively, and contained different components.

### 3.4. Determination of Total Phenolic Content and Total Flavonoid Content

#### 3.4.1. Determination of Total Phenolic Content

Total phenolic content (TPC) of the fractions was determined by the Folin-Ciocalteu (FC) method [37]. In other words, five fractions of *V. dunalianum* were dissolved in 80% methanol-water to make a 1 mg/mL solution. Then, 0.5 mL of FC reagent and 0.6 mL of sample solution were mixed as a mixture and allowed to fully react for 1 min. After that, the mixture was continued by adding 6 mL of distilled water and 1.5 mL of Na_2_CO_3_ (20% *m*/*v*) and the whole reaction system was kept in a water bath at 70 °C for 10 min. Last, it was cooled and the absorbance value was measured at 765 nm with a microplate reader by a microplate reader (Molecular Devices, Sunnyvale, CA, USA). Finally, gallic acid was used as the standard to draw the standard curve. The TPC of the samples was expressed as the amount of gallic acid (GAE) per mg of sample (µg GAE/mg extract).

#### 3.4.2. Determination of Total Flavonoid Content

The content of total flavonoids (TFC) was determined by colorimetric method [38]. A sample solution was made from five fractions to which NaNO_2_ (0.3 mL, 5% *m*/*v*) was added and allowed to stand for 8 min after mixing was completed. After that, along with 0.3 mL of Al(NO_3_)_3_ (10% *m*/*v*) was added to the mixture, and after 6 min, 2 mL of 1 M NaOH was added. After that, 0.4 mL of 70% ethanol was added. Then, the combination was held at room temperature for 12 minutes. The absorbance value at 510 nm was measured. The TFC was expressed as milligrams of rutin equivalents (RE) per gram of extract.

#### 3.4.3. Identification and Quantification of Phenolics by UHPLC-MS/MS

The five fractions of *V. dunalianum* were dissolved and filtered by mass spectrometry acetonitrile [39]. A Thermo Fisher Ultimate 3000 UHPLC system (Thermo Fisher Scientific, Bremen, Germany) equipped with a C18 column (2.1 × 100 mm, 1.9 μm, Agilent, USA) was used to analyze the main phenolic compounds in the PF, BF, CF, EF, and WF. The mobile phase consisted of acetonitrile (solvent B) and acidified ultra-pure water (0.1% acetic acid, solvent A). The gradient elution procedure was 0–3 min, 5–15% B; 3–8 min, 15–45% B; 8–15 min, 45–80% B; 15–18 min, 80–95% B; 18–20 min, 95% B. The injection volume was 2 μL, the flow rate was 0.2 mL/min, and the temperature was maintained at 30 °C. The specific parameters of the mass spectrometer are as follows: anion mode; The scanning range of MS was 100–1000 m/z; resolution, 70,000; voltage, 3.3 kV; capillary temperature, 320 °C; auxiliary flow rate, 8 L/min; and the sheath gas flow rate was 32 L/min, 4 L/min; S-lens RF level, 50%; spray voltage, 3.3 kV, capillary temperature, 320 °C; and heater temperature, 350 °C. The ions were scanned in a negative mode with a mass range from m/z 50 to 1000. The corresponding standard substances were used for the quantification of phenolic substances, and the mass spectrometry of the standard substance was obtained under the same conditions. Then, the corresponding standard curve was made according to the peak area for the quantification of phenolic substances.

### 3.5. Enzyme Inhibition Ability Assay

#### 3.5.1. Inhibitory Activity of α-Glucosidase

The experiment was carried out using a modified version of a previously described procedure [40]. The 50 μL sample was mixed with 100 µL *α-*glucosidase solution and then incubated in a constant temperature incubator at 37 °C for 15 min. Then 50 μL of 2.5 mmol/L PNPG substrate solution was added. A control was made for each group. Finally, the whole reaction system was incubated at 37 °C for 15 min. The absorbance (OD) of the reaction mixture was measured at 405 nm by a microplate reader (Molecular Devices, Sunnyvale, CA, USA).

#### 3.5.2. Inhibitory Activity of Pancreatic Lipase

The assay of inhibitory activity of five fractions on pancreatic lipase was studied by referring to the existing literature [41]. Firstly, the porcine pancreatic lipase solution was centrifuged at 4000× *g* for 5 min and the upper enzyme layer was collected. 

Secondly, the substrate solution was prepared by mixing Triton X-100 (1:100 *v*/*v*) and 5 mM sodium acetate (1:100 *m*/*v*), which needs to be left in water at 100 °C for 2 min to dissolve completely. The substrate solution was cooled before performing the relevant assay. The five fractions were dissolved in DMSO and diluted in varied quantities with Trisbuffer. As a positive control, orlistat was utilized. Finally, the substrate solution, Trisbuffer, test samples, and trypsin were mixed together and incubated for 2 h at 37 °C before centrifugation (4000× *g*, 3 min). A microplate reader was used to measure the absorbance of the reaction mixture at 400 nm. The following Equation (1) was used to compute the inhibition ratio of pancreatic lipase activity.
Inhibition (%) = [(Acontrol − Asample)/Acontrol] × 100(1)

### 3.6. Antioxidant Activity Assay

#### 3.6.1. DPPH-Radical Scavenging Capacity Assay

DPPH was carried out by the method previously described [30]. The colorimetric method was used to determine DPPH radical-scavenging ability of five fractions from *V. dunalianum* [42]. In a word, the five fractions were mixed in DPPH solution separately and mixed well, the reaction was conducted at 25 °C for 30 min. The absorbance was measured at 517 nm by using a microplate reader. All tests were performed in triplicate. The antioxidant activity of the five fractions was calculated according to the following Equation (2). Results were expressed as nmol Trolox equivalent (TE) per mg of extract (nmol TE/mg extract).
DPPH radical scavenging activity (%) = [(Acontrol − Asample/Acontrol] × 100(2)

#### 3.6.2. Determination of FRAP Reducing Ability

Ferric-reducing antioxidant power capacity was determined according to the existing literature with slight modification [43,44]. The FRAP reagent needed to be used and prepared. In short, the sample extract solution or Trolox solution was combined with freshly made FRAP reagent at 37 °C for 10 min in a dark environment. A microplate reader was used to measure the absorbance at 593 nm. Using Trolox as the benchmark, a standard curve was created, and the ferric-reducing capacity of the sample was expressed as the amount of Trolox (TE) per mg of extract (nmol TE/g extract).

#### 3.6.3. Determination of ABTS+ Free Radical Scavenging Capacity

The ABTS radical-scavenging ability of five fractions from *V dunalianum* was determined as reported in the available literature with slight modifications [45,46]. In brief, 0.5 mL of extract solution or Trolox solution was added to 4 mL of ABTS and allowed to mix thoroughly, and the mixture was reacted at 37 °C for 6 min in a light-proof environment. After reaction, the absorbance was measured at 734 nm using a microplate reader. The antioxidant activity of the five fractions was calculated according to the following equation. Calculation method is the same as DPPH.

### 3.7. Cytoprotective Effect on Oxidative Damage in H_2_O_2_-Induced HepG2 Cells

#### 3.7.1. Cell Culture and Cell Viability

Human hepatocellular carcinoma cells (HepG2) were purchased from the Kunming Cell Bank of the Chinese Academy of Sciences. The cells were cultured in a medium containing 1% penicillin, 10% fetal bovine serum and 1% streptomycin. The toxicity of the five fractions to HepG2 cells was determined using the MTT method. The steps are as follows: HepG2 cells were seeded in 96-well plates with 2.0 × 10^4^ cells per well. They were incubated in a constant temperature incubator containing 5% CO_2_ at 37 °C for 24 h, and then five fractions were added to them. After a period of time, MTT solution was added to them and incubated for 4 h. Afterwards, the MTT solution was removed, an amount of DMSO solution was added and the cells were shaken (10 min). The absorbance was recorded at 570 nm. The results showed that all five fractions were not toxic to HepG2 cells at concentrations of 50 μg/mL, 100 μg/mL, and 150 μg/mL. Therefore, the maximum concentration was selected for the experiments.

#### 3.7.2. Measurement of Intracellular Reactive Oxygen Species (ROS)

The cell treatment procedure was the same as in 3.7.1. After cell treatment, cells were collected into 1.5 mL centrifuge tubes, washed twice with 1 mL cooled PBS, collected by centrifugation at 10,000× *g* for 5 min using a 4 °C centrifuge, mixed with 1 mL of 10 μM DCFH-DA probe solution, and incubated in an incubator for 20 min. After incubation, the cells were washed twice with 1 mL of cold PBS, and then flow cytometry was used to look for Reactive Oxygen Species (ROS) (Guava easy Cyte 6-2 L flow cytometer, EMD Millipore, USA) [47].

#### 3.7.3. Determination of Cell Apoptosis

Apoptosis was measured using the human membrane-linked protein VFITC/PI apoptosis kit to determine the protective effect of each sample against H_2_O_2_-induced apoptosis in HepG2 cells. The cell treatment procedure was the same as in 3.7.1. Cells were treated with hydrogen peroxide (H_2_O_2_). Trypsin is added to the cells after they have been incubated, and once they have been detached, the cells are collected into a centrifuge tube for centrifugation, resulting in the cells being collected. After incubation, treated the cells with 100 μL of binding buffer and then treated them with membrane conjugatin V-FITC and propidium iodide (PI) for 5 min at room temperature and cold black conditions, respectively. Flow cytometry was used to investigate apoptosis right away.

#### 3.7.4. Determination of Intracellular Antioxidant Enzyme Activity

After cell treatment, cells were collected in 1.5 mL centrifuge tubes, washed twice with 1 mL of cold PBS, and collected by centrifugation at 2500× *g* for 10 min using a 4 °C centrifuge. Then 1 mL of cold PBS was added, and cell was precipitated using a 4 °C centrifuge at 2500× *g* for 10 min. The supernatant was taken from the cells to determine catalase (CAT), glutathione (GSH) and superoxide dismutase (SOD) according to the kit instructions purchased from Nanjing Jiancheng Institute of Biological Engineering.

### 3.8. Statistical Analysis

All data were expressed as Mean ± SD. Data were analyzed by one-way ANOVA of experimental data, and Tukey’s test was used to test for significant differences (*p* < 0.05). Plotting of experimental data was done using Origin 8.5 software.

## 4. Conclusions

This paper focuses on the chemical composition and biological activity of five fractions from *V. dunalianum*. The results showed that *V. dunalianum* is rich in polyphenolic chemical components. Among the five fractions, EF showed the highest TPC and TFC, and it had the best antioxidant activity, *α*-glucosidase and pancreatic lipase inhibitory actions compared to the other fractions. Moreover, all five fractions of *V. dunalianum* showed protective effects on cells in H_2_O_2_-induced oxidative stress damage. Among them, EF showed the best protective effect by inhibiting ROS production, cell apoptosis, and up-regulation of intracellular antioxidant enzyme activity (CAT, SOD, and GSH). Therefore, the bud of *V. dunalianum* can be considered an antioxidant healthy tea for treating oxidative stress-induced cell damage, and the EF fraction played an important role in activity evaluation. 

On the basis of the UHPLC-MS/MS qualitative quantitative analysis, 18 substances were detected in EF, BF, CF, but the content of compounds in the three fractions had obvious differences. These results showed that the quinic acid (**1**), chlorogenic acid (**4**) and 6’-*O*-caffeoylarbutin (**14**) were the major phenolic compounds. However, three large numbers of compounds have different content in each segment of five fractions. Specifically, the quinic acid (**1**) has the highest content in WF, about 190 times to that of EF; The chlorogenic acid (**4**) has the highest content in BF, about three, three, and five times to those of WF, EF and CF, respectively. The 6’-*O*-caffeoylarbutin (**14**) has the highest contents in EF, about 24,271 times to that of PF. According to our report [4], 6’-*O*-caffeinoyl arbutin has good antioxidant effects and enzyme inhibition effects. Therefore, the EF fraction, the highest content of 6’-*O*-caffeoylarbutin (**14**), displayed the strongest anti-oxidation effect, enzyme inhibition effect, and cytoprotective effect. In addition, other polyphenols and flavonoids may also have synergistic effects. This phenomenon clearly indicated that the use of solvents to purify and enhance phenolic compounds is a successful strategy. In summary, the research on the buds of *V. dunalianum* suggested that these could be served as an antioxidant healthy tea for treating oxidative stress-induced cell damage and as an anti-diabetic agent, which also provides a material basis for traditional applications in the food and health industry.

## Figures and Tables

**Figure 1 molecules-27-03432-f001:**
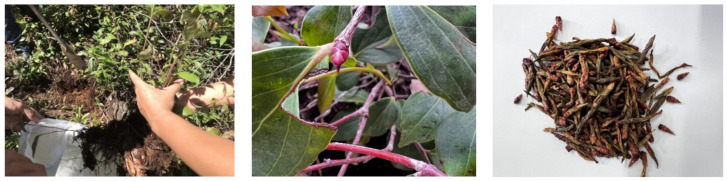
The *Vaccinium Dunalianum* Wight in nature (**Left**), enlargement with bud (**Middle**), and the “Que Zui tea” collected before extraction (**Right**).

**Figure 2 molecules-27-03432-f002:**
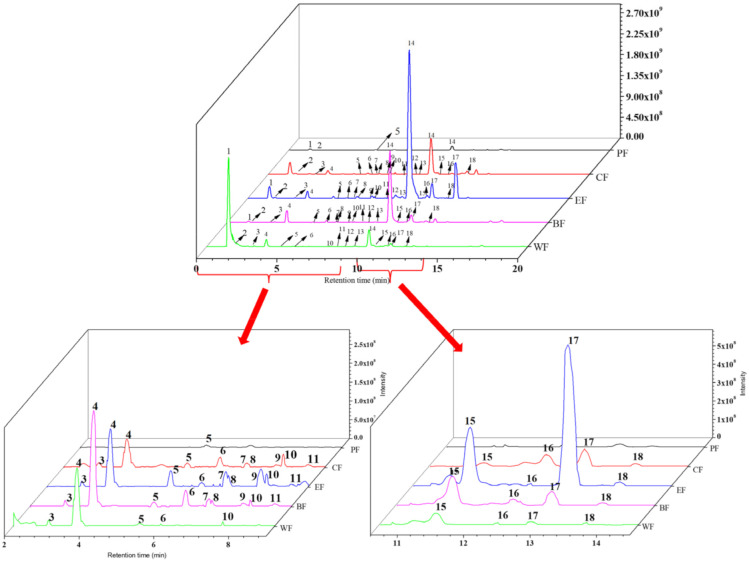
Negative ion chromatograms of phenolic compounds from PF, CF, EF, BF, and WF from *Vaccinium dunalianum.* PF, CF, EF, BF, and WF denoted petroleum ether fraction, chloroform fraction, ethyl acetate fraction, *n*-butanol fraction, and water fraction.

**Figure 3 molecules-27-03432-f003:**
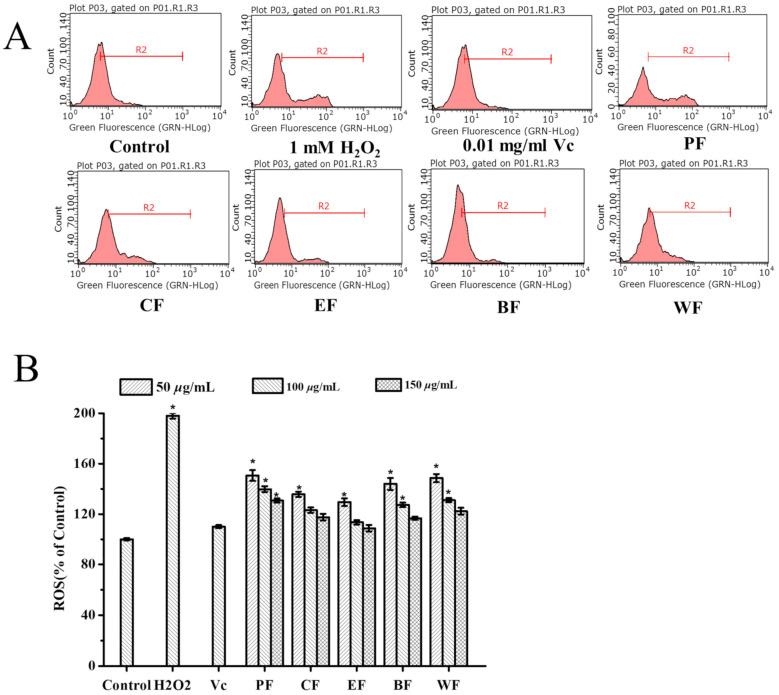
The inhibitory effects of cellular ROS generation under treatments of PF, CF, EF, BF, and WF from *V. dunalianum* in H_2_O_2_-induced HepG2 cells. (**A**) The inhibition of cellular ROS production by five fractions at the concentrations of 100 μg/mL. (**B**) The inhibition of cellular ROS production by five fractions at the concentrations of 50, 100, and 150 μg/mL. Means (bar values) with different letters are significantly different (*p* < 0.05). PF, CF, EF, BF, and WF denoted petroleum ether fraction, chloroform fraction, ethyl acetate fraction, *n*-butanol fraction, and water fraction.

**Figure 4 molecules-27-03432-f004:**
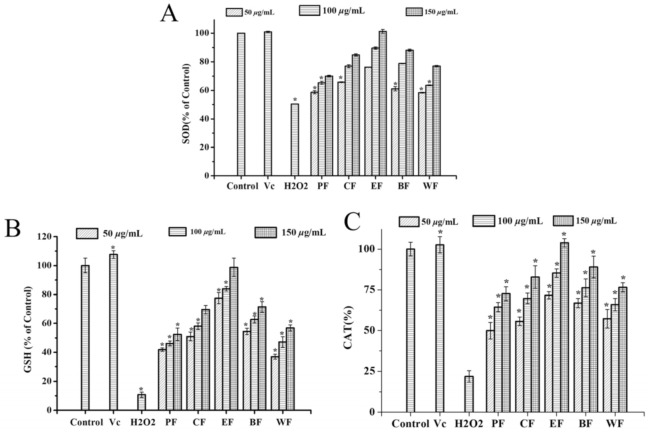
Effect of the PF, CF, EF, BF, and WF on antioxidant enzyme activity in H_2_O_2_-induced HepG-2 cells. (**A**) Superoxide dismutase (SOD) activity; (**B**) Catalase (CAT) activity; (**C**) Glutathione peroxidase (GSH-Px) activity; Means (bar values) with different letters are significantly different (*p* < 0.05). PF, CF, EF, BF, and WF denoted petroleum ether fraction, chloroform fraction, ethyl acetate fraction, *n*-butanol fraction, and water fraction.

**Figure 5 molecules-27-03432-f005:**
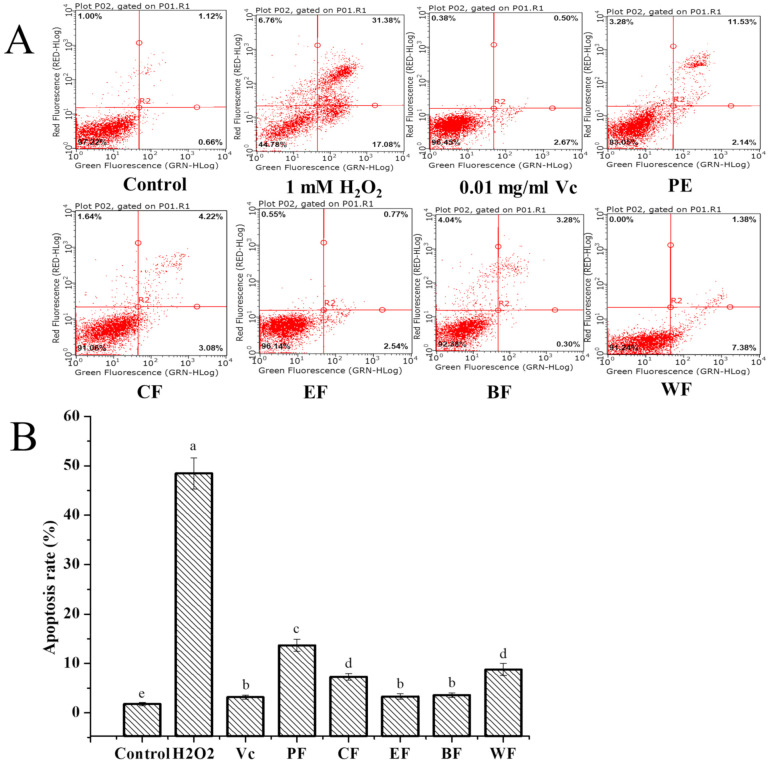
Effect of PF, CF, EF, BF, and WF from *V. dunalianum* on H_2_O_2_-induced apoptosis in HepG2 cells. (**A**) Flow cytometry analysis; (**B**) the apoptotic cell percentage of different groups. Means (bar values) differ significantly (*p* < 0.05) for different letters. PF, CF, EF, BF, and WF denoted petroleum ether fraction, chloroform fraction, ethyl acetate fraction, *n*-butanol fraction, and water fraction.

**Figure 6 molecules-27-03432-f006:**
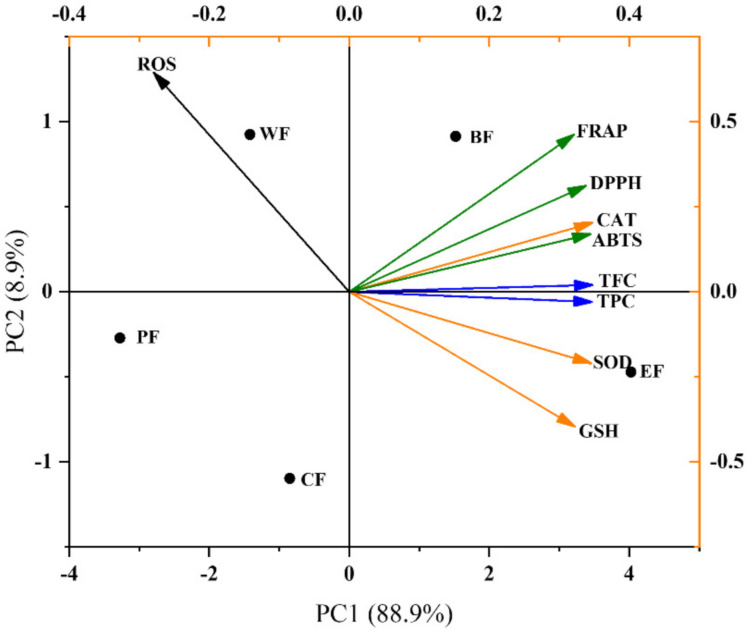
The PCA compositional analysis. Effects of PF, CF, EF, BF, and WF on total phenolics (TPC), total flavonoids (TFC), CAT, GSH, SOD, DPPH, FRAP, ABTS, and ROS. PF, CF, EF, BF, and WF denoted petroleum ether fraction, chloroform fraction, ethyl acetate fraction, *n*-butanol fraction, and water fraction. CAT, GSH, SOD, DPPH, FRAP, ABTS, and ROS denoted catalase, glutathione, superoxide dismutase, 2,2-Diphenyl-1-picrylhydrazyl, Ferric ion reducing antioxidant power, 2,2’-Azinobis-(3-ethylbenzthiazoline-6-sulphonate), and reactive oxygen species.

**Table 1 molecules-27-03432-t001:** Total phenolics content, total flavonoids content, enzyme inhibitory, and antioxidant activities of five fractions from *Vaccinium dunalianum.*

Samples	TFC ^1^	TPC ^2^	Enzyme Inhibition Activity (IC_50_ μg/mL)			Antioxidant Activity	
			α-Glucosidase	Pancreatic Lipase	DPPH (nmol TE/mg Extract) ^3^	FRAP (nmol TE/mg Extract) ^4^	ABTS (nmol TE/mg Extract) ^5^
PF	2.71 ± 0.09 ^a^	12.68 ± 0.37 ^a^	498.56 ± 12.34 ^e^	814.32 ± 6.53 ^e^	-	-	1.35 ± 0.14 ^a^
CF	9.07 ± 0.45 ^c^	21.34 ± 0.56 ^c^	201.76 ± 6.51 ^c^	253.52 ± 11.86 ^c^	0.12 ± 0.01 ^a^	0.21 ± 0.01 ^a^	5.25 ± 0.12 ^b^
EF	23.42 ± 1.10 ^e^	59.65 ± 0.93 ^e^	84.21 ± 2.21 ^a^	122.53 ± 5.45 ^a^	0.44 ± 0.03 ^d^	0.91 ± 0.04 ^d^	12.85 ± 0.23 ^e^
BF	18.79 ± 0.18 ^d^	42.09 ± 0.86 ^d^	125.55 ± 4.75 ^b^	178.76 ± 4.85 ^b^	0.35 ± 0.01 ^c^	0.84 ± 0.02 ^c^	9.26 ± 0.12 ^d^
WF	5.54 ± 0.15 ^b^	15.77 ± 0.21 ^b^	266.36 ± 1.98 ^d^	435.21 ± 8.86 ^d^	0.21 ± 0.01 ^b^	0.51 ± 0.02 ^b^	6.55 ± 0.28 ^c^

^1^ TFC: expressed as mg RE equivalent/g extract. ^2^ TPC: expressed as mg GAE equivalent/g extract. ^3^ DPPH: Expressed as nmol Trolox equivalent/mg extract. ^4^ FRAP: Expressed as nmol Trolox equivalent/mg extract. ^5^ ABTS: Expressed as nmol Trolox equivalent/mg extract. Data were from three replicates with mean ± SD; different superscript letter/s in the same column were significantly different (*p* < 0.05). PF, CF, EF, BF, and WF denoted petroleum ether fraction, chloroform fraction, ethyl acetate fraction, *n*-butanol fraction, and water fraction.

**Table 2 molecules-27-03432-t002:** Chemical composition of the five fractions from *Vaccinium dunalianum* using UHPLC- MS/MS in negative mode.

Peak	Compounds	tR (min)	[M − H]-(m/z)	Error (ppm)	Molecular Formula	MS/MS Fragment Ions	Extraction Site
**1**	quinic acid ^A^	1.17	191.0553	1.389	C_7_H_12_O_6_	85.03, 93.04, 191.06	PF, CF, EF, BF, WF
**2**	arbutin ^B^	1.51	271.0823	7.044	C_19_H_12_O_2_	86.06, 100.08	PF, CF, EF, BF, WF
**3**	6-O-caffeoyl-D- Glucopyranose ^B^	3.00	341.0877	3.024	C_15_H_18_O_9_	59.01, 135.04, 161.02, 179.03, 221.04	CF, EF, BF, WF
**4**	chlorogenic acid ^B^	3.84	353.0878	3.091	C_16_H_18_O_9_	85.03, 127.04, 191.06	CF, EF, BF, WF
**5**	caffeic acid ^A^	5.35	179.0341	1.144	C_9_H_8_O_4_	89.04, 134.04, 135.04	PF, CF, EF, BF, WF
**6**	*p*-hydroxybenzaldehyde ^A^	6.21	121.0282	1.371	C_7_H_6_O_2_	85.03, 191.06	CF, EF, BF, WF
**7**	methyl caffeate ^B^	6.83	193.0134	1.452	C_9_H_5_O_5_	91.02, 109.03, 137.02	CF, EF, BF
**8**	Ampeloptin ^C^	6.86	319.0459	3.311	C_15_H_12_O_8_	57.03, 83.01, 125.02, 137.02, 193.01	CF, EF, BF
**9**	quercetin-3-O-arabinoside ^B^	7.77	435.0906	2.868	C_20_H_19_O_11_	435.09	CF, EF, BF
**10**	5-O-(E)-Feruloylquinic acid ^C^	7.84	367.1034	2.81	C_17_H_20_O_9_	93.03, 134.03, 191.06	CF, EF, BF, WF
**11**	Neoeriocitrin ^C^	8.59	595.1671	2.291	C_27_H_32_O_15_	161.02, 323.08, 433.11	CF, EF, BF, WF
**12**	eriodictyol-7-glucoside ^C^	9.01	449.1088	2.209	C_21_H_22_O_11_	125.02, 161.02, 179.03	CF, EF, BF, WF
**13**	methyl chlorogenate ^C^	9.42	367.1067	2.564	C_17_H_20_O_9_	93.03, 134.04, 191.06	CF, EF, BF, WF
**14**	6’-O-caffeoylarbutin ^B^	10.49	433.1137	1.886	C_21_H_22_O_10_	161.02, 179.03	PF, CF, EF, BF, WF
**15**	paeoniflorin-3-O-glucoside chloride ^C^	11.17	463.1247	2.531	C_22_H_23_ClO_11_	139.04, 161.02	CF, EF, BF
**16**	*p*-hydroxybenzoic acid ^A^	12.37	137.0233	0.296	C_7_H_5_O_3_	52.03, 65.04, 93.03, 94.04	CF, EF, BF, WF
**17**	robustaside A ^B^	12.96	417.1224	2.712	C_21_H_22_O_9_	145.03, 163.04	CF, EF, BF, WF
**18**	kaempferol-3-O-β-D-glucoside ^B^	13.92	447.1396	2.453	C_21_H_20_O_11_	145.03, 160.02, 175.04, 193.05	CF, EF, BF, WF

tR: Retention time. Compound search method: ^A^: obtained from Massbank database; ^B^: obtained with standards; ^C^: obtained from references; PF, CF, EF, BF, and WF denoted petroleum ether fraction, chloroform fraction, ethyl acetate fraction, *n*-butanol fraction, and water fraction.

**Table 3 molecules-27-03432-t003:** Quantitative results of the identified and tentatively identified compounds in the five fractions from *Vaccinium dunalianum* by UHPLC- MS/MS (μg/g dry extract).

Peak	Compound	PF	CF	EF	BF	WF
**1**	quinic acid ^A^	4815.52 ± 9.85 ^b^	22,049.23 ± 49.02 ^d^	1253.7 ± 3.05 ^a^	19,221.03 ± 21.05 ^c^	239,190.25 ± 39.42 ^e^
**2**	Arbutin ^B^	62.83 ± 2.05 ^a^	1542.88 ± 3.06 ^c^	701.59 ± 7.09 ^b^	2817.86 ± 16.23 ^d^	7109.58 ± 7.65 ^e^
**3**	6-O-caffeoyl-D- Glucopyranose ^B^	-	670.24 ± 14.02 ^a^	801.55 ± 25.03 ^b^	621.75 ± 15.17 ^a^	1013.09 ± 22.06 ^c^
**4**	chlorogenic acid ^B^	-	25,730.28 ± 32.05 ^a^	42,713.84 ± 12.23 ^b^	129,305.52 ± 102.23 ^c^	47,967.83 ± 67.51 ^d^
**5**	caffeic acid ^A^	141.64 ± 2.46 ^a^	2066.85 ± 11.03 ^c^	3534.38 ± 26.85 ^d^	960.22 ± 10.41 ^b^	151.57 ± 9.51 ^a^
**6**	*p*-hydroxybenzaldehyde ^A^	-	4015.03 ± 22.92 ^c^	897.75 ± 12.67 ^b^	8736.15 ± 72.76 ^d^	88.08 ± 2.52 ^a^
**7**	methyl caffeate ^B^	-	622.65 ± 25.06 ^a^	7723.50 ± 25.03 ^c^	4014.19 ± 15.01 ^b^	-
**8**	Ampeloptin ^C^	-	1275.64 ± 26.01 ^a^	15,092.11 ± 75.63 ^c^	1711.32 ± 12.53 ^b^	-
**9**	quercetin-3-O-arabinoside ^B^	-	82.23 ± 2.51 ^a^	9451.56 ± 51.02 ^c^	899.04 ± 18.02 ^b^	-
**10**	5-O-(E)-Feruloylquinic acid ^C^	-	2207.23 ± 19.04 ^c^	2109.58 ± 10.32 ^c^	1233.83 ± 5.13 ^b^	753.55 ± 10.15 ^a^
**11**	Neoeriocitrin ^C^	-	82.23 ± 2.54 ^b^	63.96 ± 2.31 ^a^	1179.58 ± 17.64 ^d^	133.28 ± 11.06 ^c^
**12**	eriodictyol-7-glucoside ^C^	-	168.36 ± 12.74 ^b^	1470.51 ± 24.11 ^d^	554.48 ± 10.07 ^c^	54.22 ± 2.32 ^a^
**13**	methyl chlorogenate ^C^	-	361.98 ± 10.02 ^b^	6077.21 ± 17.03 ^d^	1902.92 ± 29.51 ^c^	52.01 ± 4.15 ^a^
**14**	6’-O-caffeoylarbutin ^B^	28.27 ± 2.53 ^a^	156,036.82 ± 40.07 ^c^	679,611.55 ± 17.56 ^e^	387,775.65 ± 61.99 ^d^	10,425.32 ± 54.32 ^b^
**15**	paeoniflorin-3-O-glucoside chloride ^C^	-	2379.87 ± 27.22 ^a^	20,963.06 ± 45.06 ^c^	4376.52 ± 53.03 ^b^	-
**16**	*p*-hydroxybenzoic acid ^A^	-	4608.03 ± 48.53 ^c^	86.55 ± 2.16 ^a^	1767.16 ± 29.42 ^b^	52.56 ± 12.95 ^a^
**17**	robustaside A ^B^	-	9905.46 ± 50.08 ^c^	72,950.26 ± 150.02 ^d^	7932.82 ± 62.33 ^b^	180.75 ± 23.25 ^a^
**18**	kaempferol-3-O-β-D-glucoside ^B^	-	504.39 ± 4.51 ^c^	981.48 ± 20.59 ^d^	156.46 ± 13.32 ^b^	75.15 ± 2.61 ^a^

Compound search method: ^A^: obtained from Massbank database; ^B^: obtained with standards; ^C^: obtained from references. ^a,b,c,d,e^: Represents the difference between different fractions; Different lowercase letters indicate significant differences in means within the same row (*p* < 0.05); PF, CF, EF, BF, and WF denoted petroleum ether fraction, chloroform fraction, ethyl acetate fraction, n-butanol fraction, and water fraction.

## Data Availability

Data is contained within the article.

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
