# Peer review of "Phenolic Constituents, Antioxidant and Cytoprotective Activities, Enzyme Inhibition Abilities of Five Fractions from Vaccinium dunalianum Wight"

_molecules, 2022, doi:10.3390/molecules27113432_

Round 1

Reviewer 1 Report

The authors did good work here. They are suggested to improve further the manuscript by correcting my comments. The comments are marked in yellow color.

Author Response

Reviewer1: Comments and Suggestions for Authors The authors did good work here. They are suggested to improve further the manuscript by correcting my comments. The comments are marked in yellow color (Attachement PDF).

Response: Thank you for your kind reminder. We have revised our manuscript carefully according to your and reviews comments and suggestions. We double checked English writing word for word based on your and reviewers’ comments. All the English grammar or spelling errors raised by the reviewers were revised. In addition, we have asked for Gui-Guang Cheng (one co-author), who has been published many research papers in English, polished the writing of our revised manuscript.

In the text, the sentences 1 to 4 of paragraph 2.1; the sentence 1 of paragraph 2.3.1; the sentence 1 of paragraph 2.3.3 and 2.3.3; the sentences 1 to 4 of paragraph 2.4.1; the sentences 1 to 2 of paragraph 2.4.2; the sentences 1 to 3 of paragraph 2.4.3; the review  thought that these sentences were not necessary since these are introductory words. The sentences were used to explain and describe the topic of this chapter, which help the reader to understand this paragraph of article, So the authors kept these sentences in the text. Thanks very much!

Reviewer 2 Report

The current work is about the chemical constituents of five fractions from Vaccinium dunalianum, their antioxidant activity, cytoprotective effects on H2O2-induced oxidative damage in HepG2 cells, and enzyme inhibitory effects of α-glucosidase and pancreatic lipase.

While the authors designed the experiments well and presented them well. However, their presentation, including the writing, can be further improved. Moreover, the authors really need to explain in detail the rationale for the partitioning of methanol extract into five fractions. Similarly, I think the authors need to explain why EF (Ethyl acetate fraction) has shown better results compared to other fractions. For instance, the Chemical composition analysis of five fractions (Section 2.2) shows that all EF, BF, and CF contain the same 18 chemicals, however, their activities were shown to be completely different. A detailed discussion is needed.

Author Response

Reviewer 2:

  1. The authors really need to explain in detail the rationale for the partitioning of methanol extract into five fractions.

Response: Thank you for your kind reminder. Using the difference in the solubility or partition coefficient of the substance in two mutually incompatible (or slightly soluble) solvents, the substance is transferred from one solvent to another. Therefore, the study used successively five different solvents to extract the different solubility of different components in the sample to obtain five fractions. Firstly, the 80% methanol-water extracts of buds of V. dunalianum were evaporated in vacuo to give the viscous residue extract, which was partitioned with petroleum ether to afford petroleum ether layer and H2O layer. Secondly, the H2O layer was partitioned with dichloromethane to give dichloromethane layer and H2O layer. Thirdly, the H2O layer was extracted with EtOAc, and then extracted with n-Butanol to give EtOAc layer, n-Butanol layer and H2O layer. These five fractions were partitioned with five different solvents, and contained different components.

  1. Similarly, I think the authors need to explain why EF (Ethyl acetate fraction) has shown better results compared to other fractions. For instance, the Chemical composition analysis of five fractions (Section 2.2) shows that all EF, BF, and CF contain the same 18 chemicals, however, their activities were shown to be completely different. A detailed discussion is needed.

Response: Thank you for your suggestions. The MeOH extract was successively partitioned with petroleum ether, chloroform, ethyl acetate and n-butanol to give five fractions. According to different polarity solvents, each fraction contained different components. The results showed that EF fraction had the highest total phenols (TPC) and total flavonoids (TFC), and displayed the strongest enzyme inhibition ability, antioxidant capacity (DPPH, ABTS and FRAP), and protective effect on liver cancer cells.

On the basis of UHPLC-MS/MS qualitative quantitative analysis, 18 substances were detected in EF, BF, CF, but the content of substances in the three fractions had obvious differences. The results showed that the quinic acid (1), chlorogenic acid (4) and 6'-O-caffeoylarbutin (14) were the major phenolic compounds. However, three large numbers of compounds have different content in each segment five fractions. Like the quinic acid (1) has the highest content in WF, about 190 times to that of EF; The chlorogenic acid (4) has the highest content in BF, about 3, 3, and 5 times to those of WF, EF and CF, respectively. The 6'-O-caffeoylarbutin (14) has the highest contents in EF, about 24271 times to that of PF. According to our report (Tian-Rui Zhao et.al, Molecules 2021, reference 4 in the manuscript), 6-O caffeinoyl arbutin has good antioxidant effects and enzyme inhibition effects. Therefore, the EF fraction, the highest content of 6'-O-caffeoylarbutin (14), displayed the strongest anti-oxidation effect, enzyme inhibition effect and cytoprotective effect. In addition, other polyphenols and flavonoids may also have synergistic effects. This phenomenon clearly indicated that the use of solvents to purify and enhance phenolic compounds is a successful strategy.

Reviewer 3 Report

The authors evaluate the phytochemistry and biological activity of Vaccinium dunalianum extracts obtained with different organic and aqueous solvents. In particular, the authors evaluate the content of polyphenols and flavonoids in the different solvents and test the antioxidant action. Furthermore, they evaluate the cytoprotective action on liver cells stressed with H2O2.

The manuscript is original, there are few works on the activity of Vaccinium dunalianum. However, the results are not very innovative and some critical issues need to be resolved.

It would be interesting to see some photos of the plant in nature and the samples collected before extraction. In addition, authors should include more details on the type of sample used. Did they use the leaves, stem or root? Were the grams of pulverized plant identical for all solvents? Furthermore, in what period of the year did they carry out the collection?

The authors highlight that ethyl acetate is the best solvent in the extraction processes of polyphenols and flavonoids and in the protective action. However, they do not explain in the discussion or in the conclusions the role of the different components in the biological response. In fact, is it perhaps the presence of 6-O-caffeoylarbutin or p-hydroxybenzoic acid to carry out the protective action in the EF extract? Or is it the synergistic action of the phytocomplex that determines the protective action of this extract? These aspects should also be discussed by referring to: Abate, G. et al.. Phytochemical Analysis and Anti-Inflammatory Activity of Different Ethanolic Phyto-Extracts of Artemisia annua L.. Biomolecules 2021, 11, 975. doi: 10.3390/biom11070975

Author Response

Reviewer 3:

  1. It would be interesting to see some photos of the plant in nature and the samples collected before extraction.

Response: Yes. We added the photos of the plant in nature and the samples collected before extraction.  

  1. In addition, authors should include more details on the type of sample used. Did they use the leaves, stem or root?

Response: According to our report (Shun-Hua Gao et.al, Food Chemistry 2022, reference 19 in the manuscript), the bud of V. dunalianum displayed the strongest antioxidant capacity and protective effect, compared with the leave, fruit of V. dunalianum. So the experiment was conducted using buds of Vaccinium dunalianum.

  1. Were the grams of pulverized plant identical for all solvents? Furthermore, in what period of the year did they carry out the collection?

Response: Thanks for your comment. The collected buds of V. dunalianum was dried, powder, and sieved, the dried material (2 kg) was ground into powder and sieved through 60-mesh sieve before being extracted three times with 80% methanol-water (5L) using ultrasonic aided extraction (30 min each time). And the sentence in lines 3-4 of 3.3 paragraph was corrected.

The MeOH extract was successively partitioned with petroleum ether, chloroform, ethyl acetate and n-butanol to give five fractions. And each fraction was obtained by concentration by rotary evaporation device, respectively. And we added this method in the last of 3.3 paragraph.

  1. In what period of the year did they carry out the collection?

Response: The flower buds of this sample appears in May and June of each year, and the samples were collected in mid-June. And we added in the text.

  1. The authors highlight that ethyl acetate is the best solvent in the extraction processes of polyphenols and flavonoids and in the protective action. However, they do not explain in the discussion or in the conclusions the role of the different components in the biological response. In fact, is it perhaps the presence of 6-O-caffeoylarbutin or p-hydroxybenzoic acid to carry out the protective action in the EF extract? Or is it the synergistic action of the phytocomplex that determines the protective action of this extract? Or is it the synergistic action of the phytocomplex that determines the protective action of this extract? These aspects should also be discussed by referring to: Abate, G. et al.. Phytochemical Analysis and Anti-Inflammatory Activity of Different Ethanolic Phyto-Extracts of Artemisia annua L.. Biomolecules 2021, 11, 975. doi: 10.3390/biom11070975

Response: Thanks for your suggestions. According to our report (Tian-Rui Zhao et.al, Molecules 2021, reference 4 in the manuscript) and the Reference (Abate, G. et al., Biomolecules 2021, reference 38 in the manuscript), the discussion between different components, especially 6'-O-caffeoylarbutin (14), and biological response was added in the conclusion at last of the text.

Thanks very much!

Round 2

Reviewer 3 Report

The authors made an excellent review of the manuscript. In my opinion, the manuscript can be published.